# Iron limitation of microbial phosphorus acquisition in the tropical North Atlantic

T.J. Browning[1], E.P. Achterberg[1], J.C. Yong[1], I. Rapp[1], C. Utermann[2], A. Engel[1] & C.M. Moore[3]

In certain regions of the predominantly nitrogen limited ocean, microbes can become co-limited by phosphorus. Within such regions, a proportion of the dissolved organic phosphorus pool can be accessed by microbes employing a variety of alkaline phosphatase (APase) enzymes. In contrast to the PhoA family of APases that utilize zinc as a cofactor, the recent discovery of iron as a cofactor in the more widespread PhoX and PhoD implies the potential for a biochemically dependant interplay between oceanic zinc, iron and phosphorus cycles. Here we demonstrate enhanced natural community APase activity following iron amendment within the low zinc and moderately low iron Western North Atlantic. In contrast we find no evidence for trace metal limitation of APase activity beneath the Saharan dust plume in the Eastern Atlantic. Such intermittent iron limitation of microbial phosphorus acquisition provides an additional facet in the argument for iron controlling the coupling between oceanic nitrogen and phosphorus cycles.

[1] Marine Biogeochemistry Division, GEOMAR Helmholtz Centre for Ocean Research, Kiel 24148, Germany. [2] Research Unit Marine Natural Products Chemistry, GEOMAR Helmholtz Centre for Ocean Research, Kiel 24106, Germany. [3] Ocean and Earth Science, National Oceanography Centre Southampton, University of Southampton, Southampton SO14 3ZH, UK. Correspondence and requests for materials should be addressed to T.J.B. (email: tbrowning@geomar.de).

In the (sub)tropical North Atlantic, supply of dissolved inorganic nitrogen (DIN) through diazotrophic $N_2$ fixation[1–3] results in drawdown of dissolved inorganic phosphorus (DIP), to the extent that growth of phytoplankton[1,3–5] and heterotrophic bacteria[6] can be enhanced by the simultaneous addition of both nutrients relative to supply of N alone. Microbial communities experiencing such a shortage of P upregulate production of phosphatase enzymes that hydrolyse P-esters, a large constituent of the oceanic dissolved organic P (DOP) pool, thereby supplying additional DIP for cell growth[7]. As the DOP pool can be orders of magnitude larger than that of DIP in surface ocean waters[2,8], the factors regulating access to this pool are likely important for controlling marine primary production. While the identity and prevalence of eukaryotic alkaline phosphatases (APases) remain poorly resolved[9–12], analyses of ocean metagenomic data sets have suggested that the phosphate monoesterases comprising the PhoA, PhoX and PhoD families constitute the dominant bacterial phosphatases in the ocean[13,14]. The relative availability of obligate metal cofactors activating these phosphatases have subsequently been suggested as an important environmental regulator of expression[15–17]. Until recently, cofactor requirements were understood to be two zinc (Zn) and one magnesium (Mg) ions for PhoA (ref. 18), three calcium (Ca) ions for PhoX (ref. 19) and an unknown number of Ca ions for PhoD (ref. 20). However, it is now known that PhoX and PhoD, which collectively dominate bacterial alkaline phosphatases in the ocean[13,14], have additional requirements of two and one iron (Fe) ions per enzyme, respectively[21,22]. These discoveries present the possibility that the availability of Fe, which is often depleted to biologically significant low levels in the surface ocean[23], could influence microbial P acquisition[21,22].

Eight factorial-style nutrient treatment bioassay experiments were conducted in the tropical North Atlantic in May 2015 to investigate regulation of APase by Fe and Zn supply within the context of N and P availability (Fig. 1). Four experiments in two pairs (Experiments 1 and 2 in the West Atlantic, and Experiments 4 and 5 in the East Atlantic) collectively covered the entire NPFeZn nutrient treatment factorial on each side of the Atlantic, with the remaining four individual experiments having a reduced NPFe(+NZn) design (8–9 treatment combinations in triplicate). Surface seawater collection and treatment was conducted using strict trace-metal-clean procedures and incubated water was assessed after ∼48 h for changes in chlorophyll $a$ concentrations, flow cytometric cell counts, and APase activities (APA) measured via a fluorometric assay[24]. Treatment responses were interpreted relative to triplicate unaltered control bottles, and the phytoplankton community composition, nutrient and trace metal concentrations of initial seawater. We find that Fe can become limiting to whole microbial community APase activity distal to the Saharan dust plume in the tropical Western North Atlantic. Such a role for Fe in P acquisition, alongside requirements for both photosynthesis and $N_2$ fixation[2–4], further underscores the critical role of this element in regulating the productivity of past and future (sub)tropical ocean systems.

## Results

### Cross-Atlantic nutrient trends.
As for previous studies in the (sub)tropical North Atlantic, we encountered waters that were depleted in DIP (mean excess DIP (DIP*) = 9.9 nmol l$^{-1}$, s.d. = 16.0 nmol l$^{-1}$, $n = 37$) yet host to a relatively large residual DOP pool (mean = 322 nmol l$^{-1}$, s.d. = 72 nmol l$^{-1}$, $n = 33$; Fig. 2)[2,8]. Cross-basin measurements of dissolved Fe displayed concentrations that were generally enhanced in the East Atlantic (0.46 to 2.37 nmol l$^{-1}$ east of 30 °W) relative to the west (0.24 to 1.56 nmol l$^{-1}$ west of 30 °W), consistent with

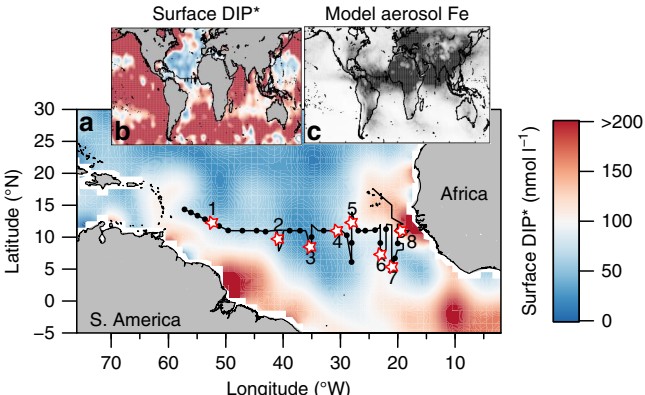

**Figure 1 | Experiment locations in context of cross-Atlantic nutrient trends.** (**a**) Experimental start locations (stars) on a background of surface ocean DIP* calculated from annual average WOA nutrient concentrations. DIP* represents excess available dissolved inorganic phosphorus (DIP) based on a 16:1 N:P requirement of phytoplankton ( = DIP–DIN/16; DIN is dissolved inorganic nitrogen). The thin black line is the cruise transect. (**b**) DIP* as for the main panel but at a global scale highlighting DIP deficiency in the North Atlantic. (**c**) Total soluble aerosol iron (Fe) deposition estimated by a model (relative units, darker shading = higher Fe deposition).

greater aerosol deposition on this side of the Atlantic (Fig. 2)[25,26]. This trend was clearer for dissolved Mn, for which variability in the open low latitude surface ocean is typically dominated by aerosol flux[27]. Dissolved Zn concentrations were generally >5–10 times lower than Fe apart from around 40–45 °W. Elevated concentrations of all three metals at the far western edge of the cruise track likely originated from shelf and/or Amazon River-derived inputs[27].

### Microbial responses to nutrient amendment.
Changes in chlorophyll $a$ concentrations and cell counts in the experiments confirmed a general pattern of N proximally limiting the overall phytoplankton community, alongside potential NP co-limitation in some circumstances (Fig. 3)[5]. Specifically, increases in mean chlorophyll $a$ concentrations in N-treated bottles were not significantly different from control bottles in Experiments 1, 4, and 7, but were with supplementary addition of P (and occasionally Fe or Zn) implying conditions of, or approaching, community NP co-limitation[5,16]. Clear proximal and secondary limitation by N and P, respectively, were observed in Experiments 3 and 6, whereby addition of N produced a significant chlorophyll $a$ increase above any treatment without N, and N + P further enhanced this. Finally, no evidence of secondary P limitation was observed in Experiment 8 in waters associated with the East Atlantic Mauritanian upwelling, which had the least P-deficient starting seawater (Figs 1 and 2). These community-level changes in chlorophyll $a$ were also largely reflected in cell counts and cellular fluorescence of *Prochlorococcus*, *Synechococcus* and photosynthetic eukaryotes (Supplementary Figs 1–4). Although less clear than for phytoplankton, counts of heterotrophic bacteria also suggested NP co-limitation in Experiments 1, 4 and 7 (Supplementary Fig. 5)[6].

Addition of DIP often resulted in suppression of average APA (Experiments 1, 3–6), consistent with downregulation of APase expression and/or degradation of pre-existing APase enzymes upon supply of readily accessible P (Fig. 4). Addition of inorganic N in the absence of DIP enhanced APA in Experiments 1, 3, 6 and 7 (for example, N-treated APAs were 1.75–2.6 times higher than

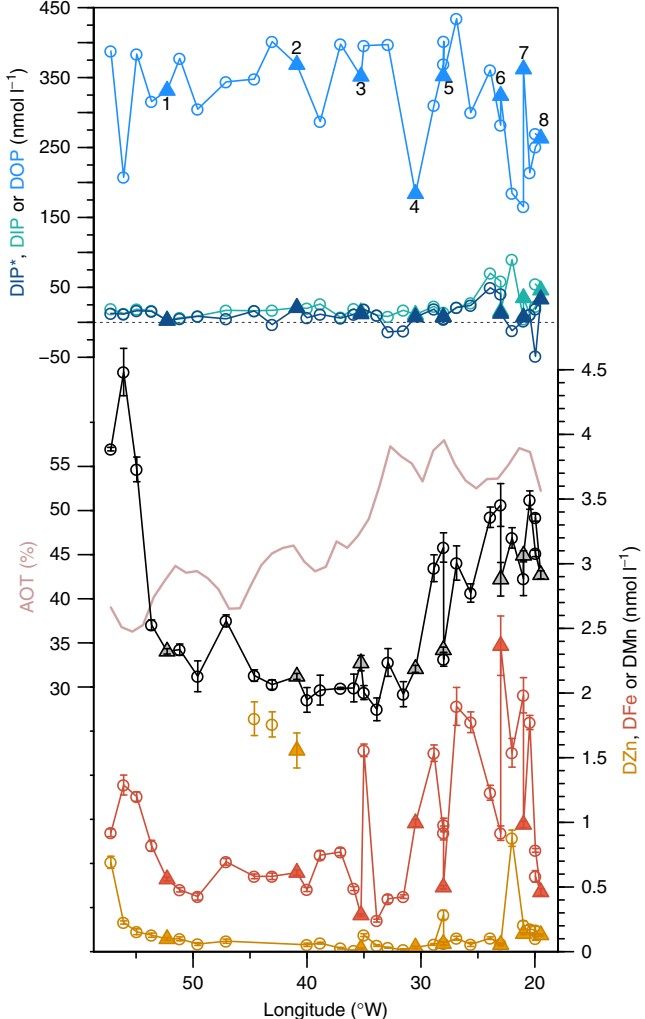

**Figure 2 | Cross-Atlantic nutrient trends.** DIP = dissolved inorganic phosphorus, DIP* represents excess available DIP based on a 16:1 N:P requirement of phytoplankton ( = DIP–DIN/16; DIN is dissolved iorganic nitrogen), DOP = dissolved organic phosphorus, DFe = dissolved Fe, DZn = dissolved zinc, and DMn = dissolved manganese, AOT = aerosol optical thickness (satellite-derived average for May 2015). Error bars on metal concentrations represent the s.d. calculated from the precision of the isotope ratios and counting of the mass spectrometer. Sampling locations are highlighted with black dots in Fig. 1a and filled triangles indicate concentrations measured at experiment start points. High dissolved Zn concentrations around 40–45 °W indicate potential trace Zn contamination of these three samples.

controls in these experiments). These experiments appeared to be under conditions where phytoplankton approached NP co-limitation, and the responses were consistent with artificial relief of N (co-)limitation further enhancing the degree of P stress and hence driving up-regulation of APA. We observed no cases where treatment with Zn, alone or in combination with N, enhanced APA; however, Fe treatment appeared to play a role in enhancing APA in all experiments in the West Atlantic (Experiments 1–3; Fig. 4). Most significantly, in Experiment 2 the Fe treatment alone induced significant APA increases, without artificially enhancing P stress via coincident N addition (Fig. 4b; Fe-amended APA was 7.2 times higher than the control).

Three key interpretations can be drawn from our observations. First, these experiments have demonstrated the potential for

a biogeochemical dependence of P acquisition on Fe availability, with the underlying mechanistic link presumably being the requirement for Fe in activating the most widespread bacterial APase enzymes in the ocean, PhoX and PhoD[13,14,21,22]. Second, they have demonstrated that APA was restricted by the availability of Fe in some regions of the tropical North Atlantic at the time of sampling (Fig. 4b), despite all experiments—even those in the western part of the basin—being in a sector of the North Atlantic with generally elevated Fe supply from aerosols (Fig. 1c). Third, while acknowledging the potential for differential sensitivity to Zn and Fe limitation within our experiments due to contrasting cellular localizations of different APases[14] (see Supplementary Discussion), in contrast to rates measured in Zn-amended treatments within previous experiments employing similar methods in the North Atlantic[15], our experiments provided limited evidence in tested seawaters for P–Zn or P–Fe–Zn interactions.

## Discussion

Our findings are relevant for understanding the coupling and evolution of nutrient cycles and their response to anthropogenic drivers (Supplementary Fig. 6). The essential role of Fe in marine $N_2$ fixation is well known[3,4], and for this reason Fe has been suggested to regulate N input to the ocean, thus indirectly exerting a control on oceanic productivity over long timescales[28]. Ultimately the oceanic P reservoir sets an upper constraint on productivity over geological timescales, as unlike N, inputs to the surface ocean are not biologically regulated[29]. However, the extent to which the upper productivity constraint imposed by the P inventory is approached will be dictated by the overall efficiency of upper ocean P utilization over both short[30] and long[28] timescales. Any restriction of overall DOP reactivity due to trace metal requirements could thus be hypothesized to represent one mechanism for reducing overall P utilization efficiency.

As for the photosynthetic apparatus and $N_2$ fixation, APases likely first evolved in an ancient ocean enriched in Fe and lower in Zn (ref. 17). A maintained reliance on PhoX and PhoD, for microbes[13,14], in the modern well-oxygenated ocean may initially appear paradoxical given the extent of Fe depletion in many surface regions[23]. However, it is reasonable to assume that broad-scale marine biogeochemical systems where microbial communities had a tendency to experience P stress in the geological past likely resulted from elevated $N_2$ fixation depleting excess DIP. This in turn would require an adequate and sustained Fe supply, as is the case in the modern day (sub)tropical North Atlantic[3]. Making these assumptions, it thus follows that only under relatively Fe-enriched conditions would a strong dependence on APases exist—potentially reducing selective pressure away from the use of Fe-dependant PhoX and PhoD. However, on smaller geographic scales, for example across marked nutrient province boundaries[3], strong and dynamic micro- and macro-nutrient gradients[31] may potentially impose selection pressures conferring advantages to APase enzymes with alternative cofactors (for example, ref. 32).

In contrast to Fe, enhanced aerosol deposition is not synonymous with elevated Zn (Fig. 2). Furthermore Zn appears to be highly depleted both in the surface (Fig. 2) and more generally within the upper Atlantic ocean[33,34]. This is partly due to the depletion of this element within the source waters feeding the thermocline, particularly of the Atlantic, as a result of high removal within the Southern Ocean[33–35]. Both limited aerosol sources and high removal rates from the upper ocean at high latitudes in the Southern Ocean could hence lead to relatively low Zn availability within anticipated regions of DIP depletion within high Fe, low latitude systems, further pointing

towards reduced selective advantage for widespread Zn-containing PhoA. Recognition of Fe as a common cofactor for both $N_2$ fixation and dominant DOP hydrolysis enzymes may thus help reconcile the high $N_2$ fixation rates measured in regions of the (sub)tropical North Atlantic with DIP deficiency (Fig. 1b,c)[3,36]. However, alongside previous observations[15], our data suggest that the relatively short residence time of Fe in the surface waters of this region[37], combined with variable

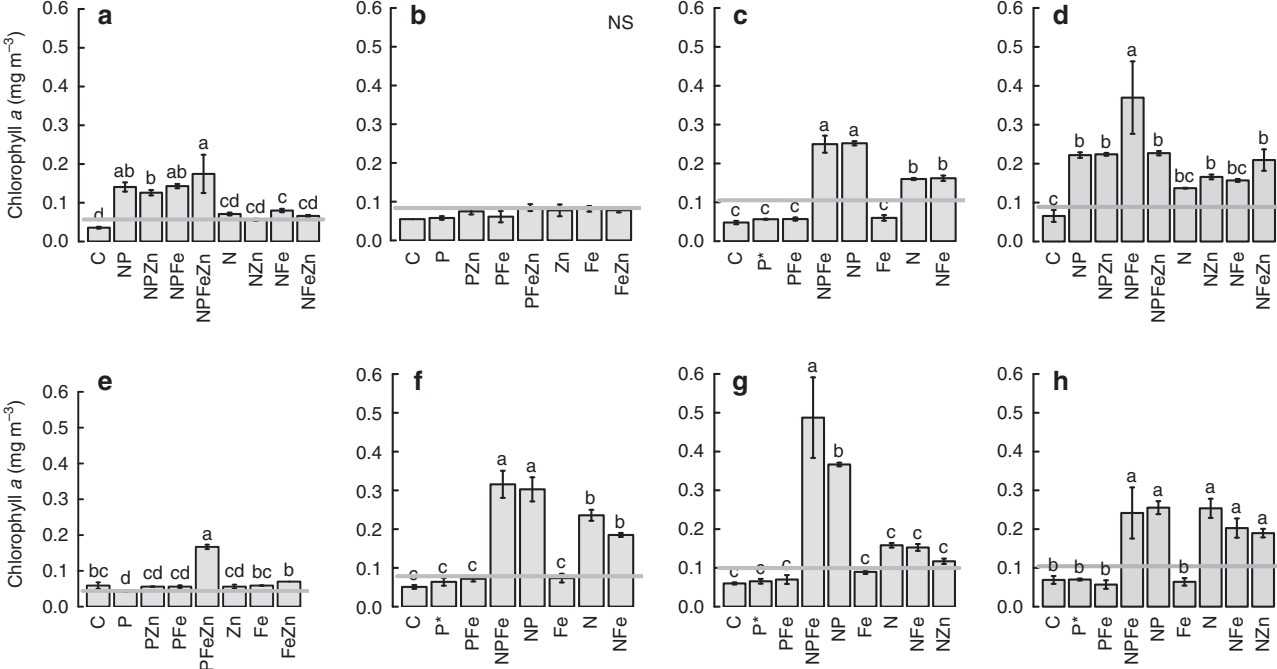

**Figure 3 | Chlorophyll *a* responses to nutrient additions.** (**a**–**h**) Experiments 1–8. Shown are means ± s.e.'s, $n = 3$ for all except where indicated by an asterisk ($n = 2$). The grey horizontal line is the mean initial time point measurement ($n = 3$). Treatment means were compared using a one-way analysis of variance and a Fisher PLSD means comparison test (indistinguishable means labelled with the same letter ($P < 0.05$); NS is 'not significant').

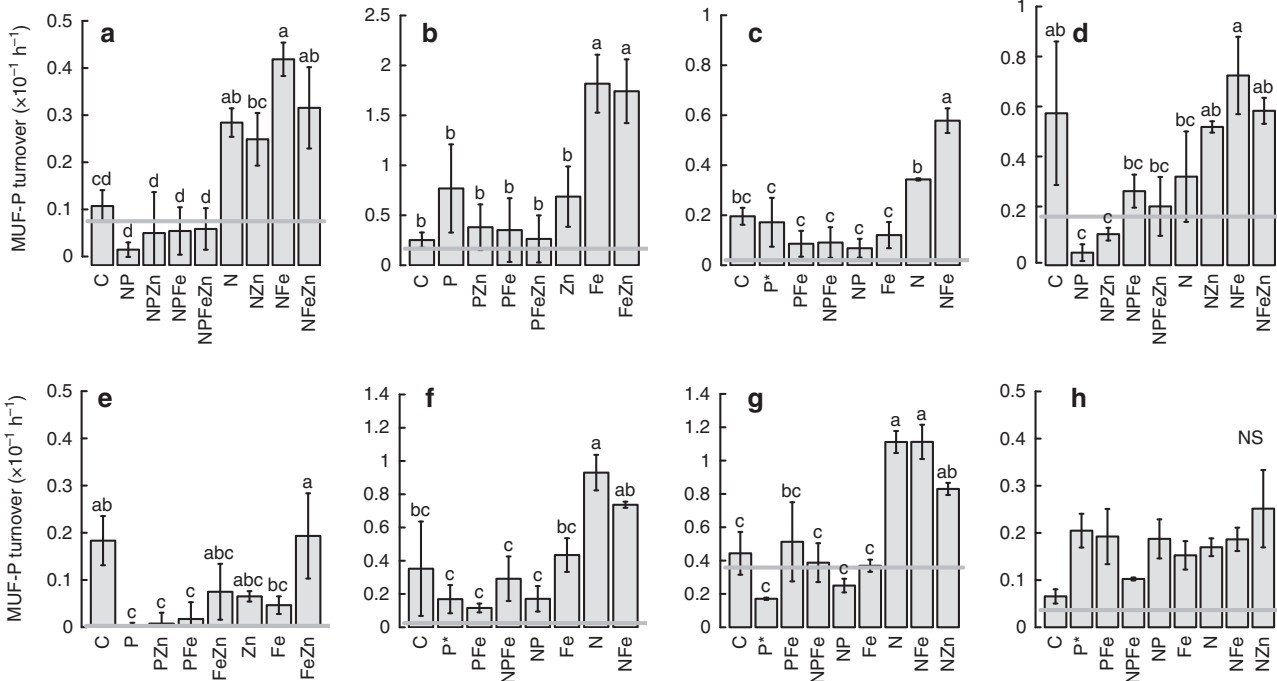

**Figure 4 | Alkaline phosphatase activity responses to nutrient additions.** (**a**–**h**) Experiments 1–8. Activities are shown as turnover rates of the spiked DOP substrate, 4-methylumbelliferyl phosphate (MUF-P). Shown are means ± s.e.'s, $n = 3$ for all except where indicated by an asterisk ($n = 2$). The grey horizontal line is the mean initial time point measurement ($n = 3$). Treatment means were compared using a one-way ANOVA and a Fisher PLSD means comparison test (indistinguishable means labelled with the same letter ($P < 0.05$); NS is 'not significant').

Fe inputs[31], may both result in spatiotemporal variability in DOP requirements and hence at least localized incidences where Fe and/or Zn (ref. 15) becomes limiting for APA (for example, Fig. 4b).

Although the consequences of such intermittent trace metal limitation of APA remain difficult to quantify for the overall use efficiency of P at a system scale, anthropogenic perturbations have the potential to further alter any interactions between the Zn, Fe and P cycles. Anthropogenic N inputs to the oceans continue to increase rapidly[38,39]. Resultant feedbacks remain poorly constrained, with one possibility being a displacement of $N_2$ fixation that may leave P demand largely unaltered[1,40]; while alternatively, if anthropogenic N input becomes additive to $N_2$ fixation and coupled with a lower limit in the plasticity of P requirements of phytoplankton communities[41,42], a shift to proximal P limitation could be induced[39]. In the latter case, our findings would point towards any coincident alterations to fluxes and bioavailability of Fe, and possibly Zn (ref. 15), becoming increasingly important in regulating DOP utilization and hence potentially the P use efficiency and productivity of (sub)tropical oceanic systems. Future work adding important detail to the role of Fe demonstrated in this study, including the potential specificity of different APases for diverse DOP quality[7,14,43] and developing linkages between APase expression, abundance and activity for the most relevant taxonomic groups[43–46], will help in developing the foundation needed to address these questions.

## Methods

**Research cruise and seawater collection.** Experiments and sample collection were carried out on the RV Meteor M116 cruise (2 May 2015 to 3 June 2015). Surface (approximately 2–3 m depth) seawater was sampled from a custom-built towed-fish fitted with acid-washed tubing and suction provided by a Teflon bellows pump. Water was pumped directly into a purpose-built clean air laboratory container with positive air pressure maintained via inward airflow passed through a HEPA filter. All sample collection and nutrient spiking was carried out in the clean lab container.

**Nutrient amendment experiments.** Approximately 48 h duration on-deck incubation experiments were carried out in 1 or 4 l trace-metal-clean Nalgene polycarbonate bottles. Seawater was collected at local night using the trace-metal-clean towed-fish described previously. Filling times were ~40 min for 1 l bottle experiments (total volume = 27 l) and ~2 h for 4 l bottle experiments (108–120 l depending on experiment). The bottle filling protocol followed that of similar experiments (for example, ref. 47) to distribute total seawater collected throughout the filling period across all bottles, with initial (that is, $t = 0$) bottles being filled at the beginning, middle, and end of the filling time period. Bottled seawater was spiked with one of the following combinations of nutrients/trace metals: N, P, Fe, NP, NFe, PFe, NPFe (1 l bottle experiments; Experiments 3 and 6); N, P, Fe, NP, NFe, PFe, NPFe, NZn (1 l bottle experiments; Experiments 7 and 8); N, NP, NFe, NZn, NPFe, NPZn, NFeZn, NPFeZn (4 l bottle experiments; Experiments 1 and 4); P, Fe, Zn, PFe, PZn, FeZn, PFeZn (4 l bottle experiments; Experiments 2 and 5).

Triplicate control bottles with no nutrients added were also collected and incubated alongside all treatment experiments. Treated bottles were spiked to the following nutrient/trace metal concentrations: N = 1 µmol l$^{-1}$ NO$_3^-$ + 1 µmol l$^{-1}$ NH$_4^+$; P = 0.2 µmol l$^{-1}$ PO$_4^{3-}$; Fe = 2 nmol l$^{-1}$ Fe$^{3+}$ (as FeCl$_3$ in 2% HCl); Zn = 2 nmol l$^{-1}$ Zn$^{2+}$ (as ZnCl$_2$ in 2% HCl). N and P treatment solutions were previously passed through a prepared Chelex 100 column to remove trace metal contamination. Bottles were placed in on-deck incubators connected to the ships underway flow-through system to continuously maintain temperatures at that of sea surface waters. Incubators were screened with Blue Lagoon screening (Lee Filters), which maintained irradiance at ~30% of that of the surface. After ~48 h incubation, experiments were taken down and subsampled for chlorophyll *a* concentrations, flow cytometry and APA.

**Chlorophyll *a*.** A volume of 500 ml samples were filtered onto Macherey Nagel GFF filters and extracted for 12–24 h in 10 ml 90% acetone in a −20 °C freezer in the dark before measurement on a Turner Designs trilogy fluorometer following the method of Welschmeyer (1994)[48]. Fluorometrically determined concentrations were corrected systematically to high-performance liquid chromatography (HPLC)-derived chlorophyll *a* concentrations using a linear correlation between duplicate samples taken on the M116 cruise that were measured by both methods ($r^2 = 0.79$, $P < 0.001$; $n = 135$).

**Alkaline phosphatase activities.** Time course analyses of alkaline phosphatase activity were conducted with 20 ml subsamples of initial/incubated seawater using 100 nmol l$^{-1}$ 4-methylumbelliferyl phosphate (MUF-P; Sigma-Aldrich) as the organic phosphate substrate and directly following the protocol of Ammerman (1993)[24]. For bacterial cells, the APA assay substrate (MUF-P) is accessible to APase enzymes located within the periplasm outwards[49], collectively encompassing the majority of PhoX and PhoA, and ~50% of PhoD, predicted through analysis of the Global Ocean Sampling metagenomic database[14]. Fluorescence was measured on a Turner Designs Trilogy fluorometer equipped with a custom 355/10 nm (excitation) – 460/10 nm (emission–detection) snap-in module manufactured by Turner Designs. Following MUF-P spiking, fluorescence measurements were performed at $t = 0$, 1.5, and 3 h. APA (h$^{-1}$) was calculated as fluorescence of 100 nmol l$^{-1}$ 4-methylumbelliferone (MUF; Sigma-Aldrich) divided by the initial ($t = 0$ to $t = 1.5$ h) slope of fluorescence time course (fluorescence per hour). Assays were performed at night (i.e., ~48 h time point) and incubated in the dark submerged in continuously running seawater from the ships underway system. To our knowledge, there is no evidence of diel variability in APA and performing the experiments at night in the dark eliminates fluctuations in one environmental variable. MUF-P turnover rates were calculated using the fluorescence gradient between 0 and 1.5 h (that is initial slope) as recommended by Ammerman (1993)[24], although using the gradient between 0 and 3 h yielded similar results. Regular Milli-Q blanks and paraformaldehyde-killed controls were conducted and generally yielded fluorescence values similar to $t = 0$ readings. As we employed this method as a relative comparison of APA between experimental treatments, using water taken from the same site, in this instance we largely circumvent any potential limitations of following the (widely used) practice of employing a single MUF-P concentration[50].

**Trace metal concentrations.** Samples were collected in acid-washed 125 ml LDPE sample bottles for dissolved (0.8/0.2 µm AcroPack1000 filter capsule) trace metal concentrations (metals: Fe, Zn, Mn, Mg, Cu, Co, Cd, Pb and Ni). Samples were acidified with concentrated ultra-pure hydrochloric acid to pH 1.9 under a laminar flow hood within 1–2 days of collection. Samples were measured via pre-concentration on a seaFAST system (Elemental Scientific Inc.) and subsequent analysis using an Element XR inductively coupled plasma-mass spectrometer (ICP-MS) following the method of Milne *et al.* (2010)[51]. Isotope dilution was used for measurement of Fe, Zn, Mg, Cu, Cd and Ni and standard additions were used to quantify Mn, Co, and Pb (ref. 50). Analysed concentrations were validated via good agreement with SAFe certified reference material.

**Nutrient concentrations.** Samples were collected in acid washed polypropylene tubes for low concentration analyses of dissolved inorganic nitrate + nitrite and phosphate (15 ml) and DOP (50 ml). Both were frozen immediately in a −30 °C freezer before subsequent transfer to a −20 °C freezer. Both DOP and dissolved inorganic N and P samples were filtered (0.8/0.2 µm AcroPack1000 filter capsule) as recommended by Patey *et al.* (2010)[52]. Low concentration dissolved inorganic nitrate + nitrite and phosphate samples were analysed on return to a land-based laboratory using a custom-built analyser equipped with 2 m waveguides (World Precision Instruments Inc.) as described by Patey *et al.* (2008)[53]. Total dissolved phosphate (TDP) samples were digested under elevated pressure (1.5 bar) and temperature (120 °C) for 30 min following addition of potassium peroxide, sodium hydroxide and boric acid, and then analysed using a SEAL QuAAtro nutrient autoanalyser system (SEAL Analytical). Samples were blank-corrected for P content of digestion chemicals. DOP concentration was subsequently calculated as DOP = TDP − DIP. TDP determined by this method results in near complete conversion of DOP to DIP and thus leads to higher measured DOP concentrations than widely used UV digestion methods (which liberate on the order of ~70% DOP (ref. 54)). Results for both analytical methods were checked against certified reference material distributed by KANSO Technos, Japan.

**Flow cytometry.** Flow cytometry samples (2 ml) were fixed with neutralized paraformaldehyde at a 1% final concentration, vortex-mixed and left in the dark for 10 min before transfer to a −80 °C freezer. Upon return to a regular laboratory, samples were thawed at room temperature and analysed using a FACSCalibur flow cytometer (Becton Dickenson, Oxford, UK). Cell counts were carried out in CellQuest software (Becton Dickenson). Plots of orange fluorescence versus red fluorescence were used to identify and enumerate *Synechococcus* from other picophytoplankton, and plots of side scatter versus red fluorescence (with *Synechococcus* gated out) were used to enumerate photosynthetic picoeukaryotes and *Prochlorococcus*. Heterotrophic bacteria were identified and enumerated from a separate nucleic acid stained (SYBR Green) aliquot using plots of side scatter versus green fluorescence with cyanobacteria gated out.

**Aerosol optical thickness.** The aerosol optical thickness (AOT) line presented in Fig. 2 is 1° resolution Moderate Resolution Imaging Spectroradiometer (MODIS) May 2015 AOT from NASA (http://neo.sci.gsfc.nasa.gov/), latitude-averaged over 5.5 to 14.5 °N and smoothed with a 3-point moving average.

**Model aerosol Fe distribution.** In Fig. 1c, the map of modelled total soluble aerosol Fe deposition (combustion + desert dust sources) was made using data presented in Luo *et al.* (2008)[55].

**Data availability.** Data sets for this study are available from the corresponding author upon request.

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

## Acknowledgements

We thank the captain, crew, and principle scientists (M. Visbeck and T. Tanhua) of the RV Meteor M116 cruise. B. Berks, J. LaRoche, J.S. Tolman, J. Zorz, E. Bertrand, C. Löscher, H. McClelland, C. Schlosser, and M. Patey are thanked for useful discussion, and J. Pampin-Baro, P. Streu, K. Nachtigall, and T. Klüver for technical support. N. Mahowald is thanked for providing the model Fe aerosol dataset in Fig. 1. This work was funded by a Marie Skłodowska-Curie Postdoctoral European Fellowship (OceanLiNES; Grant number 658035) and supporting Future Ocean Excellence cluster grant awarded to T.J.B. The research cruise was funded by the Deutsche Forschungsgemeinschaft as part of Sonderforschungsbereich 754 'Climate-Biogeochemistry Interactions in the Tropical Ocean'. We thank three reviewers for their valuable comments on the manuscript.

## Author contributions

T.J.B., C.M.M. and E.P.A. designed the research. T.J.B. led the study. T.J.B. and C.U. performed the experiments at sea. I.R. and T.J.B. analysed the seawater trace metal concentrations. J.C.Y. and T.J.B. performed the nutrient concentration analysis. A.E. oversaw the flow cytometry analysis. T.J.B. wrote the initial draft of the paper and T.J.B., C.M.M. and E.P.A. edited subsequent drafts. All authors commented on the manuscript.

## Additional information

**Competing interests:** The authors declare no competing financial interests.

