## [Peer Review File · Nature Communications]

Reviewers' Comments:

Reviewer #1 (Remarks to the Author)

Browning et al. present a very interesting hypothesis, namely that iron limits microbial utilization of organic phosphorus via the control of iron-dependent alkaline phosphatase enzyme activity. They test this hypothesis with a set of comprehensive field experiments at a series of hydrostations in the tropical North Atlantic. They show enhanced alkaline phosphatase activity following iron amendments within low-iron regions. From these results they conclude that iron limitation controls the utilization of the complex, largely uncharacterized pool of dissolved organic phosphorus (DOP), and possibly N₂ fixation and productivity. I really enjoyed reading this paper. It is both timely and important.

I have just three comments/questions:

1. The authors claim that the iron-dependent PhoX and PhoD forms of alkaline phosphatase are the most important, yet these were not measured in the study. How do they know that these enzymes were present?
2. The authors propose that iron stimulates the enzyme that controls DOP hydrolysis, yet DOP hydrolysis was not measured in this study. Does an increase in in vitro enzyme activity translate into a more rapid turnover of DOP? At the very least, it would have been nice to see an estimate of alkaline phosphatase reactive substrate concentrations, or a measurement of the drawdown of DOP during the shipboard incubations.
3. The authors added iron in the form of acidified iron chloride, yet in nature there is probably little, if any, free iron in most marine ecosystems. It is currently believed that most (all) iron is complexed to organic ligands and that the ligands are responsible for iron transport into the cell. If this is true, then what is the sequence, kinetics and dynamics between free iron addition, ligand binding, transport and stimulation of iron-dependent enzymes? Is the putative PhoX/PhoD inside or outside of the cell (most alkaline phosphatase in marine systems is extracellular)?

Minor issues:

1. Are the PhoX/PhoD annotations robust? Have the proteins been isolated and characterized?
2. It is ironic that such an elegant and comprehensive experiment uses "total chlorophyll a" as the primary indicator of enhanced growth. What about heterotrophic organisms?
3. Figure 1: I would delete the surface "DIP*" since this is a derived, not a measured, value. The WOCE program did not employ high sensitivity methods for nanomolar nutrient concentrations.
4. Figure 2 is an important figure, but in its present format is too small to convey the results (the format of Figure S2, not in the main text, is much better).
5. Supplemental information: the first two paragraphs read "like an apology" and could/should be removed.
6. Figure S3 caption: <0.2 um seems wrong
7. Figure S4: Not much is said about the relationship between increases in fluorescence per cell and increases in cell number (Figures S1 and S4).

8. Some of the data for cell number increases following metal treatments are “many-fold” following just a 48-hr incubation. These would imply very large “net” growth rates, some larger than laboratory measured maximum growth rates under nutrient saturated conditions.

Reviewer #2 (Remarks to the Author)

This study poses an interesting question: Does trace metal availability modulate the phosphorus cycle by determining the extent of dissolved organic phosphorus (DOP) utilization? The idea is based on the fact that different alkaline phosphatase enzymes require different metal cofactors. To explore this question, the authors conducted a series of bottle incubation experiments along a transect through the N Atlantic Ocean that transitioned from low to high dust deposition areas. They then measured the alkaline phosphatase activity after 48hr using the MUF-P protocol and concluded that Fe availability limits DOP hydrolysis in low dust areas, while the response is repressed in high dust areas.

My main concern with the study stems from the strengths of the conclusions that can be drawn based on the MUF-P assay. The authors discuss in the supplement, but not the main text, that the ability of the MUF-P substrate to enter the cell could yield different activity estimates for PhoX, PhoA, and PhoD, because the latter two enzymes are thought to be located in the cytoplasm, while the former is periplasmic. This matters because if the MUF-P is not readily taken into the cell, or if the rate of uptake is slower than the rate of hydrolysis, then the contribution from PhoA and PhoD could be underestimated. By extension, it also means that the method might not be sensitive enough to assess the role of zinc in enhancing DOP hydrolysis, because the Zn-requiring PhoA enzyme is cytoplasmic. To address this question, additional metrics to verify the results for Zn are needed. For example, were any of the experimental trends confirmed with an additional substrate (eg PNP, which may have yielded different rates if it more easily enters the cell), or was gene expression of the three enzymes measured to confirm that PhoA was not responsive? Even if only available for a subset of the experiments, without these independent verifications, I do not think it is possible to draw conclusions about if, how, or where zinc availability influences the phosphorus cycle.

Figure 1B would be improved if the scale for DIP* is blown up. Also, could [DIP] be added to this chart? It would be useful to see the actual data in addition to the calculated DIP* values.

Paragraph beginning line 112: This paragraph poses an interesting argument – that anywhere with enough Fe for N fixation would likely also have enough Fe to support PhoX and PhoD expression, thereby relaxing the selective pressure to rely on other metal cofactors for P acquisition. However, this seems like an oversimplification. While it may hold in regions like the high Fe eastern Atlantic, the ocean is full of transitional zones where nutrient gradients cause cells to transition from replete to limited status, sometimes over very small distances. In those locations, there would still be selective pressure to use AlkPases with different metal cofactors. Additionally, non-diazotrophs would not need to make this optimization “choice” between AlkPase and nitrogenase, so the effect on their AlkPases metal content would presumably be more direct. I would like to see these caveats discussed a bit more thoroughly in the paragraph, because it has very big implications for understanding the biogeographical distributions of cells (and competition between diazotrophs and others).

Paragraph beginning line 56: It would be appropriate to reference some of the recent studies in this region that show similar trace metal trends across the basin.

Reviewer #3 (Remarks to the Author)

Overview:

This study uses factorial-style nutrient amendment bioassays to show that iron availability has a

strong effect in phosphorus (P) acquisition from dissolved organic P compounds in the tropical North Atlantic. Iron has been identified as a main factor controlling nitrogen (N) fixation in this oceanic region, and nitrogen fixers usually up-regulate the production of phosphohydrolases to acquire P from organic compounds. The authors bring another element into the picture by demonstrating that Fe additions stimulate the activity of these enzymes, pointing to a control of Fe on the oceanic N and P cycles. I find the results compelling but I think there are some important aspects that should be addressed in the discussion, as detailed below.

The rationale behind the experiments is that out of the three dominant phosphatases of the ocean: PhoA, PhoD, and PhoX, it has been recently discovered that the more widespread PhoD and PhoX require Fe as a cofactor, in contrast to the traditional PhoA, which requires Zn. Until now, Zn was believed to control P acquisition from dissolved P compounds (phosphoesters) by marine phytoplankton.

General comments:

One of the first things that came to my mind when reading the ms is that the study seems mostly focused on phytoplankton, while the phosphatases PhoD and PhoX have been mostly found in heterotrophic bacteria. It is true that cyanobacteria like *Trichodesmium*, *Synechococcus* and some uncultured *Prochlorococcus* strains harbour PhoX, but the authors should take into account that Fe addition may be also activating heterotrophic bacterial phosphatases, especially in this oceanic area, where P may limit heterotrophic bacterial activity (Cotner et al. 1992). Did the authors measure some kind of bacterial activity in the experiments? It would have been a nice addition to the story.

In eukaryotes, PhoX has been only been identified in *Volvox* and the freshwater chlorophyte *Chlamydomonas reinhardtii*. However, as the ms is now written it seems that also eukaryotic phosphatases are encoded by these genes, whereas the genes coding for most eukaryotic phosphatases have not been yet identified, and their cofactors are largely unknown. This observation does not change the outcome of the study, because even if heterotrophic bacteria account for a large portion of the P cycling the conclusions are still valid from a community point of view. But I strongly suggest the authors include this discussion in the paper.

It is also possible that the phosphatases of phytoplankton inhabiting this area of the ocean use Fe as cofactor, but as I said the cofactors for eukaryotes are largely unknown.

Specific comments:

Line 10 and line 68: I would erase the "fixed" and just say limitation by N. I find the word "fixed" confusing. Same in line 140.

Line 95: in line with what I mentioned before, PhoX and PhoD are abundant and widespread yes, but in bacteria. This is an important piece of information.

Lines 99-101. Zn did not have an effect from a community point of view, but it did have in *Synechococcus* cells. Some *Synechococcus* strains harbour both PhoX and PhoA (Cox and Saito 2013), so maybe that's the reason that both Zn and Fe stimulated *Synechococcus* cells (e.g Figure S1 exp 2 and exp 5).

Figure S4. This figure is not mentioned in the text, and does not add any important information.

Figure S5. I think this figure should be in the main text.

Responses to reviewers' comments

Manuscript: 'Iron limitation of microbial phosphorus acquisition in the tropical North Atlantic' by T.J. Browning et al.

Responses to comments are in bold.

Reviewer #1 (Remarks to the Author):

Browning et al. present a very interesting hypothesis, namely that iron limits microbial utilization of organic phosphorus via the control of iron-dependent alkaline phosphatase enzyme activity. They test this hypothesis with a set of comprehensive field experiments at a series of hydrostations in the tropical North Atlantic. They show enhanced alkaline phosphatase activity following iron amendments within low-iron regions. From these results they conclude that iron limitation controls the utilization of the complex, largely uncharacterized pool of dissolved organic phosphorus (DOP), and possibly N₂ fixation and productivity. I really enjoyed reading this paper. It is both timely and important.

We are very pleased the Reviewer enjoyed reading the paper and found it both a timely and interesting topic. We thank them for the time taken to review our manuscript and discuss each of their comments sequentially below.

I have just three comments/questions:

1. The authors claim that the iron-dependent PhoX and PhoD forms of alkaline phosphatase are the most important, yet these were not measured in the study. How do they know that these enzymes were present?

As the Reviewer suggests, we infer that PhoX and PhoD forms of alkaline phosphatase are generally more abundant in the ocean than PhoA. This is based on the findings of two bioinformatics studies (Luo et al., 2009; Sebastian and Ammerman et al., 2009), which identified numbers of PhoX/D/A homologues in the Global Ocean Sampling metagenomic database (hosting data from samples collected on a 8000 km transect through the North Atlantic and Equatorial Pacific oceans; See Rusch et al., 2007). The studies collectively found PhoX/D to be more abundant than PhoA. Such findings have been supported by more recent studies (e.g. Kathuria and Martiny, 2011).

In light of this comment and the suggestions made by Reviewer 3, we have amended our manuscript to clarify that the presence of PhoX/D/A is inferred on the basis of analyses of these large-scale metagenomic programmes.

Starting line 31:

“Whilst the identity and prevalence of eukaryotic alkaline phosphatases (APases) remain poorly resolved^{9,10}, analyses of ocean metagenomic datasets have suggested

that the phosphate monoesterases comprising the PhoA, PhoX, and PhoD families constitute the dominant bacterial phosphatases in the ocean^{11,12}.”

2. The authors propose that iron stimulates the enzyme that controls DOP hydrolysis, yet DOP hydrolysis was not measured in this study. Does an increase in in vitro enzyme activity translate into a more rapid turnover of DOP? At the very least, it would have been nice to see an estimate of alkaline phosphatase reactive substrate concentrations, or a measurement of the drawdown of DOP during the shipboard incubations.

The alkaline phosphatase assay we employed measures hydrolysis of the artificially added DOP substrate, MUF-P. We are directly monitoring its hydrolysis in the alkaline phosphatase assay (i.e. it is DOP hydrolysis we are measuring as a quantitative indicator of enzyme activity). Drawdown of DOP in the experiments was not measured, as this would have required offline trace-nutrient clean filtration methods for collection of high numbers of filtrates, which was unfortunately not practically feasible in our experimental campaign.

3. The authors added iron in the form of acidified iron chloride, yet in nature there is probably little, if any, free iron in most marine ecosystems. It is currently believed that most (all) iron is complexed to organic ligands and that the ligands are responsible for iron transport into the cell. If this is true, then what is the sequence, kinetics and dynamics between free iron addition, ligand binding, transport and stimulation of iron-dependent enzymes? Is the putative PhoX/PhoD inside or outside of the cell (most alkaline phosphatase in marine systems is extracellular)?

We argue that although questions of iron speciation, cycling and uptake may inherently be of scientific interest, they are not directly relevant to any of the interpretation or conclusions that can be drawn from experiments of the type performed. However, for completeness we agree that, as the reviewer points out, virtually all dissolved iron in seawater is ligand-bound, which acts to keep it in solution. Within our experiments, it is likely that a significant fraction of the free iron supplied will be rapidly complexed by excess natural organic ligands already present in the seawater (e.g., Buck et al., 2015 measured iron-binding ligands in excess of dissolved Fe in all sampled surface waters in a cross-basin North Atlantic cruise). Subsequent acquisition of either an enhanced free iron pool or the enhanced ligand bound pool can then occur through the same variety of mechanisms which are currently thought to contribute to uptake within natural systems (see e.g. Shaked and Lis 2012; Lis et al. 2015). However, within our experiments the key point is that in these experiments we are adding a pool of excess iron. The fact that some of this is accessed is evident from the induced biological responses. The mechanisms of access are not directly relevant.

Minor issues:

1. Are the PhoX/PhoD annotations robust? Have the proteins been isolated and characterized?

PhoX has been isolated/characterized (e.g., Quisel et al., 1996; Hallman, 1999; Moseley et al., 2006; Wu et al., 2007) and putative PhoX has been identified/characterized in some marine cyanobacteria (Orchard et al., 2009; Kathuria and Martiny, 2011). PhoD is the least well characterized out of PhoX/D/A but has still been isolated/characterized (Eder et al., 1996; Kageyama et al., 2011). As Reviewer 3 points out it may well be the case that other alkaline phosphatases are more widespread than currently resolved. Because PhoX/D/A are reported in the literature as dominant marine alkaline phosphatases (e.g., Luo et al., 2009; Sebastian and Ammerman et al., 2009; Kathuria and Martiny, 2011) we focused our experiments around their known metal cofactors (i.e., iron and zinc) and framed our discussion of bulk community APA changes around potential regulation of these enzymes in particular. We refer to ‘main comment #1’ describing how our revised text accentuates uncertainty over exactly the types of alkaline phosphatases contributing to the bulk community alkaline phosphatase activity we measured.

2. *It is ironic that such an elegant and comprehensive experiment uses “total chlorophyll a” as the primary indicator of enhanced growth. What about heterotrophic organisms?*

We use total chlorophyll-a as it gives a good qualitative indication of bulk community phytoplankton changes, is easily measured on-ship using a sensitive fluorescence technique, and is what other similar types of studies have used previously (e.g., Mills et al., 2004; Moore et al., 2008). Flow cytometry cell counts of cyanobacteria were given in the original supplementary information and generally show similar changes to chlorophyll-a biomass. Although not presented in the original manuscript, we also performed flow cytometry analysis of SYBR Green stained aliquots for enumeration of heterotrophic bacteria (i.e., with cyanobacteria gated out). These counts have been added as a new figure in the revised Supplementary Information (new Supplementary Fig. 5). Although responses to nutrient amendment are less clear than for phytoplankton they do show some evidence for NP co-limitation, as has been previously observed (Mills et al., 2008). This new figure is referred to in the revised manuscript.

Starting line 83 the revised manuscript now reads:

“These community-level changes in chlorophyll-a were also largely reflected in cell counts and cellular fluorescence of *Prochlorococcus*, *Synechococcus* and photosynthetic eukaryotes (Supplementary Figs 1-4). Although less clear than for phytoplankton, counts of heterotrophic bacteria also suggested NP co-limitation in Experiments 1, 4, and 7 (Supplementary Fig. 5)⁶.”

3. *Figure 1: I would delete the surface “DIP*” since this is a derived, not a measured, value. The WOCE program did not employ high sensitivity methods for nanomolar nutrient concentrations.*

We agree with the Reviewer that the WOCE programme did not employ nanomolar nutrient concentration analyses. Consequently, absolute concentrations of the

derived quantity DIP* will have some inaccuracies at very low levels ($\sim < 50\text{nM}$). However the large scale gradients presented in Figure 1 are robust to such caveats (see e.g., Deutsch et al. 2007; Moore et al. 2009) and the key point is to highlight the apparent exceptional nature of the (sub)tropical North Atlantic with regards to the relatively lower surface measured P concentrations compared to N at a global ocean scale. For this reason we consider the inclusion of the discussed subpanel to provide crucial context.

4. *Figure 2 is an important figure, but in its present format is too small to convey the results (the format of Figure S2, not in the main text, is much better).*

Our aim with this figure was to have both chlorophyll-a and APA for any given experiment in the same column (i.e., to facilitate comparison of the two datasets). We have now reformatted the figure in the revised manuscript so that it is split over more rows, allowing the figure to be enlarged.

5. *Supplemental information: the first two paragraphs read “like an apology” and could/should be removed.*

Based on one of Reviewer 2’s comments we have amended the first paragraph of supplementary information. Moreover some of the content from this section has been moved to the revised Methods section and concluding paragraph.

6. *Figure S3 caption: $< 0.2\ \mu\text{m}$ seems wrong*

This was a typographical error and should have read $2\ \mu\text{m}$. We thank the reviewer alerting us to this.

7. *Figure S4: Not much is said about the relationship between increases in fluorescence per cell and increases in cell number (Figures S1 and S4).*

We have amended our discussion of the results to briefly note that both cell counts of cyanobacteria and cellular fluorescence generally follow changes in bulk chlorophyll-a concentrations.

Starting on line 83 of the revised manuscript the amended sentence reads:

“These community-level changes in chlorophyll-a were also largely reflected in cell counts and cellular fluorescence of *Prochlorococcus*, *Synechococcus* and photosynthetic eukaryotes (Supplementary Figs 1-4).”

8. *Some of the data for cell number increases following metal treatments are “many-fold” following just a 48-hr incubation. These would imply very large “net” growth rates, some larger than laboratory measured maximum growth rates under nutrient saturated conditions.*

Although high, our maximum calculated net growth rates within nutrient amended bottles (e.g. 1.3 d^{-1} for *Synechococcus* in N+P amended bottles in Experiment 7) are well within the range for maximum growth rates under nutrient saturated growth. Moreover, these are similar to results found in similar types of bioassay experiments conducted in the tropical North Atlantic (Davey et al., 2008), and presumably reflect high maximal growth rates of extant communities that are reached under these warm, sunlit conditions upon nutrient enrichment.

Reviewer #2 (Remarks to the Author):

This study poses an interesting question: Does trace metal availability modulate the phosphorus cycle by determining the extent of dissolved organic phosphorus (DOP) utilization? The idea is based on the fact that different alkaline phosphatase enzymes require different metal cofactors. To explore this question, the authors conducted a series of bottle incubation experiments along a transect through the N Atlantic Ocean that transitioned from low to high dust deposition areas. They then measured the alkaline phosphatase activity after 48hr using the MUF-P protocol and concluded that Fe availability limits DOP hydrolysis in low dust areas, while the response is repressed in high dust areas.

My main concern with the study stems from the strengths of the conclusions that can be drawn based on the MUF-P assay. The authors discuss in the supplement, but not the main text, that the ability of the MUF-P substrate to enter the cell could yield different activity estimates for PhoX, PhoA, and PhoD, because the latter two enzymes are thought to be located in the cytoplasm, while the former is periplasmic. This matters because if the MUF-P is not readily taken into the cell, or if the rate of uptake is slower than the rate of hydrolysis, then the contribution from PhoA and PhoD could be underestimated. By extension, it also means that the method might not be sensitive enough to assess the role of zinc in enhancing DOP hydrolysis, because the Zn-requiring PhoA enzyme is cytoplasmic. To address this question, additional metrics to verify the results for Zn are needed. For example, were any of the experimental trends confirmed with an additional substrate (eg PNP, which may have yielded different rates if it more easily enters the cell), or was gene expression of the three enzymes measured to confirm that PhoA was not responsive? Even if only available for a subset of the experiments, without these independent verifications, I do not think it is possible to draw conclusions about if, how, or where zinc availability influences the phosphorus cycle.

The Reviewer is calling for an independent verification of our alkaline phosphatase activity results for zinc treatments, with their reasoning being that PhoA is located in the cytoplasm. Whilst respecting the Reviewers concern, we think this statement not fully consistent with the available literature. Moreover, although we acknowledge that any caveats with respect to cellular localization may be relevant with regards to interpretation in the context of specific enzymes, our overall experimental design, and specifically our primary conclusion, is fully robust to any such caveats. Below we outline our arguments, but in acknowledging the reviewers

point, which we initially highlighted ourselves within the supplemental material, we have amended both the main manuscript and the supplementary material.

We assume Reviewer 2 is referring to the computational/bioinformatics results of Luo et al. (2009), as originally discussed within our supplementary paragraph 1, because as far as we are aware there is no experimental evidence for cytoplasmic localization of PhoX/A/D. Regardless, from their bioinformatics results, Luo et al. report that around 60% of PhoA enzymes are actually *non-cytoplasmic*, and are instead either located extracellularly, in the periplasm, or are membrane-associated – all locations that are currently assumed to be accessible to the supplied MUF-P (Martinez and Azam, 1993). Any substantial PhoA up-regulation resulting from Zn addition would therefore still be expected to result in a significant corresponding enhancement of MUF-P hydrolysis and consequently, assuming above background noise, there is no reason that such a response shouldn't be observable within experiments of the type performed. Indeed we note that Mahaffey et al. (2014) reported MUF-P resolved enhanced APA following Zn amendment within some very similar experiments.

Beyond the challenges of biomass-intensive genomic or proteomic characterization of potential PhoX/D/A shifts within individual treatments, such independent expression studies would encounter difficulties in generating a causative link with our measured bulk seawater APA rates. Most natural PhoX/D/A genes/enzymes are not characterized and the available sequenced AP genes are reported to be highly variable amongst microbes (Lin et al., 2012); thus any study undertaking this challenge must first perform sequencing of extant populations in the seawater and identify putative PhoX/D/A genes for sufficiently large numbers of taxonomic groups of bacteria (and potentially eukaryotes) before becoming representative of the bulk, community-level APA measured in our assays. Beyond this formidable task, it also remains possible that any measured expression parameter is purely P-regulated, or even by other factors such as the diurnal cycle, and therefore may have little correspondence to actual activities of alkaline phosphatase enzymes (which in turn might depend on numbers of alkaline phosphatase protein structures and their activation by a metal such as Fe or Zn).

To surmise, we very much agree with the Reviewer that in general more research activities should be undertaken to resolve the cellular location of alkaline phosphatase enzymes and their relative abilities to access the spectrum of natural and artificial DOP substrates, alongside starting the task of genetically characterizing natural AP enzymes and controls on their expression and activity (which may differ). We have added such recommendations to the main text of our revised manuscript. Both of these suggestions for future work, however, are beyond the immediate aims and scope of our study. We reiterate that currently there is no direct evidence to suggest that PhoA would be inhibited from hydrolyzing MUF-P to the extent that it would not be registered in our assays and as a result stand firm with regards to our interpretations of our zinc amendments. Moreover, and more significantly, we also reiterate that our key conclusion and the major novelty within

the study regards the clear demonstration that enhanced Fe availability significantly enhanced APA.

Revisions pertinent to these points are made in the following places within the revised manuscript:

Starting line 107:

“Thirdly, while acknowledging the potential for differential sensitivity to Zn and Fe limitation within our experiments due to contrasting cellular localizations of different APAases¹² (see Supplementary Discussion), in contrast to rates measured in Zn-amended treatments within previous experiments employing similar methods in the North Atlantic, our experiments provided limited evidence in tested seawaters for P-Zn or P-Fe-Zn interactions.”

Starting line 164 (concluding paragraph):

“Future work adding important detail to the role of Fe demonstrated in this study, including the potential specificity of different APases for diverse DOP quality^{7,12,40} and developing linkages between APase expression, abundance and activity for the most relevant taxonomic groups⁴⁰⁻⁴³, will help in developing the foundation needed to address these questions.”

Starting line 214 (Methods – note that in the revised manuscript all methods are in the main manuscript text):

“Time course analyses of alkaline phosphatase activity were conducted with 20 mL subsamples of initial/incubated seawater using 100 nmol L⁻¹ 4-methylumbelliferyl phosphate (MUF-P) (Sigma-Aldrich) as the organic phosphate substrate and directly following the protocol of Ammerman (1993)²². For bacterial cells, the APA assay substrate (MUF-P) is accessible to APase enzymes located within the periplasm outwards⁴⁶, collectively encompassing the majority of PhoX and PhoA, and ~50% of PhoD, predicted through analysis of the Global Ocean Sampling metagenomic database¹².”

Within the Supplementary Information (whole edited section reproduced below):

Subcellular production and operation of alkaline phosphatase: implications for community phosphate and fluorometric APA measurement

Computational prediction of the subcellular location of PhoX, PhoD and PhoA from marine bacterial metagenomes suggest a more dominant extracellular or periplasmic localization of PhoX APases, with <20% cytoplasmic localization¹. In contrast, the same study predicted >40% of PhoA and PhoD marine bacterial APases to be more cytoplasmic in localization¹. Consequently, in addition to potential involvement in any internal cycling², a significant fraction of PhoA and

PhoD activity in situ may preferentially depend on DOP transported through cell membranes into the cytoplasm¹.

Any tendency for cytoplasmic localization of PhoA and PhoD could bias fluorometric APA assays using substrates that might not be transported across cell membranes (e.g. MUF-P)¹. Noting that ~60% of PhoA and PhoD APases are likely still in a localization that could be accessible to such substrates, there is thus at least the possibility that our experimental design may underestimate the potential for Zn limitation of overall DOP hydrolysis rates within the community. Specifically, under any conditions where Zn was initially limiting the synthesis of PhoA, it might be expected that the MUF-P derived APase response to subsequent Zn amendment within our experiments may be suppressed below any total increase in hydrolysis. We further note that within some experiments we found APA in NFeZn treated bottles to be lower than that of NFe treated bottles, and in some cases NZn lower than N alone. Although equivocal, one potential explanation for such results could be a shift towards reliance on community level P acquisition through enhanced intracellular transport to support increased cytoplasmic PhoA activity following increased Zn supply. Subsequently, extracellular/periplasmic hydrolysis as measured through the MUF-P APA assay could be hypothesised to be relatively suppressed in such circumstances. However, given that >50% of PhoA APases might still be expected to respond, we would still expect to be able to resolve some significant response within such an experimental design, as demonstrated in previous studies³.

The potential for differential localization of Zn binding PhoA (and Fe binding PhoD) relative to Fe binding PhoX described above would ultimately lead to our experimental protocol being less sensitive to potential Zn limitation than Fe limitation. Consequently, although the lack of evidence for Zn limitation of APase activity should be treated with some caution, we note that our primary conclusion, that community level APA can be restricted by Fe availability within sub-regions of our study area, is fully robust to any such caveats. Moreover, the extracellular/periplasmic dominance of PhoX localization relative to PhoD would suggest the former is more likely to be driving the responses we observed. Additionally, it is likely that the PhoX family of enzymes act on a broader range of DOP substrates (i.e. both those that can and cannot be transported through the cell membrane) and hence any increased activity of Fe dependent PhoX is more likely to be linked to enhanced community level access to extracellular DOP through liberated DIP¹. Such a mechanism may be of particular benefit for organisms within defined ecological niches in the modern (sub)tropical ocean, including the microbiome represented by *Trichodesmium* sp. colonies⁴.

Figure 1B would be improved if the scale for DIP is blown up. Also, could [DIP] be added to this chart? It would be useful to see the actual data in addition to the calculated DIP* values.*

With our original scale we were aiming to enable quantitative comparison of DOP and DIP pools. Taking the Reviewers comment on board we have simply assigned

additional space to the macronutrient section of the figure, to both enable better resolution of cross-basin changes in the inorganic pool whilst maintaining ready comparison between sizes of the inorganic and organic pools. The DIP concentrations have also now been added to the plot.

Paragraph beginning line 112: This paragraph poses an interesting argument – that anywhere with enough Fe for N fixation would likely also have enough Fe to support PhoX and PhoD expression, thereby relaxing the selective pressure to rely on other metal cofactors for P acquisition. However, this seems like an oversimplification. While it may hold in regions like the high Fe eastern Atlantic, the ocean is full of transitional zones where nutrient gradients cause cells to transition from replete to limited status, sometimes over very small distances. In those locations, there would still be selective pressure to use AlkPases with different metal cofactors. Additionally, non-diazotrophs would not need to make this optimization “choice” between AlkPase and nitrogenase, so the effect on their AlkPases metal content would presumably be more direct. I would like to see these caveats discussed a bit more thoroughly in the paragraph, because it has very big implications for understanding the biogeographical distributions of cells (and competition between diazotrophs and others).

We are glad the reviewer found this an interesting and thought provoking idea. Clearly this is a conceptual simplification with relevance to broad scale selection pressures over evolutionary timescales. We have now stressed this in the revised manuscript and followed the Reviewer’s advice in including a more thorough discussion of caveats.

Starting line 125 the revised paragraph now reads:

“As for the photosynthetic apparatus and N₂ fixation, APases likely first evolved in an ancient ocean enriched in Fe and lower in Zn (ref. 15). A maintained reliance on PhoX and PhoD, for microbes^{11,12}, in the modern well-oxygenated ocean may initially appear paradoxical given the extent of Fe depletion in many surface regions²¹. However, it is reasonable to assume that broad-scale marine biogeochemical systems where microbial communities had a tendency to experience P stress in the geological past likely resulted from elevated N₂ fixation depleting excess DIP. This in turn would require an adequate and sustained Fe supply, as is the case in the modern day (sub)tropical North Atlantic³. Making these assumptions, it thus follows that only under relatively Fe enriched conditions would a strong dependence on APases exist – potentially reducing selective pressure away from use of Fe-dependant PhoX and PhoD. However, on smaller geographic scales, for example across marked nutrient province boundaries³, strong and dynamic micro- and macro-nutrient gradients³⁴ may potentially impose selection pressures conferring advantages to APase enzymes with alternative cofactors (e.g. ref. 29).”

Paragraph beginning line 56: It would be appropriate to reference some of the recent studies in this region that show similar trace metal trends across the basin.

Following the Reviewers suggestion we have now added references of studies that show similar trace metal trends across the basin.

In addition to Rijkenberg et al. (2014) (reference #25), we now also refer to Rijkenberg et al. (2012) (reference #23) and Conway and John (2014) (reference #24).

Reviewer #3 (Remarks to the Author):

Overview:

This study uses factorial-style nutrient amendment bioassays to show that iron availability has a strong effect in phosphorus (P) acquisition from dissolved organic P compounds in the tropical North Atlantic. Iron has been identified as a main factor controlling nitrogen (N) fixation in this oceanic region, and nitrogen fixers usually up-regulate the production of phosphohydrolases to acquire P from organic compounds. The authors bring another element into the picture by demonstrating that Fe additions stimulate the activity of these enzymes, pointing to a control of Fe on the oceanic N and P cycles. I find the results compelling but I think there are some important aspects that should be addressed in the discussion, as detailed below.

We were very pleased to read that the Reviewer found the results of our study compelling. We thank the Reviewer for their valuable input and refer to detailed responses to all of their comments below.

The rationale behind the experiments is that out of the three dominant phosphatases of the ocean: PhoA, PhoD, and PhoX, it has been recently discovered that the more widespread PhoD and PhoX require Fe as a cofactor, in contrast to the traditional PhoA, which requires Zn. Until now, Zn was believed to control P acquisition from dissolved P compounds (phosphoesters) by marine phytoplankton.

General comments:

One of the first things that came to my mind when reading the ms is that the study seems mostly focused on phytoplankton, while the phosphatases PhoD and PhoX have been mostly found in heterotrophic bacteria. It is true that cyanobacteria like Trichodesmium, Synechococcus and some uncultured Prochlorococcus strains harbour PhoX, but the authors should take into account that Fe addition may be also activating heterotrophic bacterial phosphatases, especially in this oceanic area, where P may limit heterotrophic bacterial activity (Cotner et al. 1992). Did the authors measure some kind of bacterial activity in the experiments? It would have been a nice addition to the story.

We agree with the Reviewer concerning the focus of the initial manuscript on phytoplankton, rather than all marine microbes. Although we note that prokaryotes dominate the autotrophic population within our study region, we thank the reviewer

for highlighting that we cannot exclude the influence of heterotrophic bacteria. We have now modified the manuscript text in multiple locations to take this into account.

Although not presented in the original manuscript, we also performed flow cytometry analysis of SYBR Green stained aliquots for enumeration of heterotrophic bacteria (with cyanobacteria gated out). These counts have been added as a new figure in the revised Supplementary Information (new Supplementary Fig. 5). Although responses to nutrient amendment are less clear than for phytoplankton they do show some evidence for NP co-limitation, as has been previously observed (Mills et al., 2008). This is referred to in the revised manuscript. We note that within our experiments our aim was replicated measurements of community-level APA. As such we did not measure any bacteria-specific APA, although agree that results from such rate measurements would have been an interesting addition.

Specific changes in the revised manuscript are:

Starting line 23:

“In the (sub)tropical North Atlantic, supply of dissolved inorganic nitrogen (DIN) through diazotrophic N₂ fixation¹⁻³ results in drawdown of dissolved inorganic phosphorus (DIP), to the extent that growth of phytoplankton^{1,3-5} and heterotrophic bacteria⁶ can be enhanced by the simultaneous addition of both nutrients relative to supply of N alone.”

Starting line 31:

“Whilst the identity and prevalence of eukaryotic alkaline phosphatases (APases) remain poorly resolved^{9,10}, analyses of ocean metagenomic datasets have suggested that the phosphate monoesterases comprising the PhoA, PhoX, and PhoD families constitute the dominant bacterial phosphatases in the ocean^{11,12}.”

Starting line 39:

“However, it is now known that PhoX and PhoD, which collectively dominate bacterial APases in the ocean^{11,12}...”

Starting line 85:

“Although less clear than for phytoplankton, counts of heterotrophic bacteria also suggested NP co-limitation in Experiments 1, 4, and 7 (Supplementary Fig. 5)⁶.”

Starting line 102:

“...with the underlying mechanistic link presumably being the requirement for Fe in activating the most widespread bacterial APase enzymes in the ocean, PhoX and PhoD^{11,12,19,20}.”

Starting line 125:

“As for the photosynthetic apparatus and N₂ fixation, alkaline phosphatases likely first evolved in an ancient ocean enriched in Fe and lower in Zn (ref. 15). A maintained reliance on PhoX and PhoD, for microbes^{11,12}, in the modern well-oxygenated ocean may initially appear paradoxical given the extent of Fe depletion in many surface regions²¹.”

In eukaryotes, PhoX has been only been identified in Volvox and the freshwater chlorophyte Chlamydomonas reinhardtii. However, as the ms is now written it seems that also eukaryotic phosphatases are encoded by these genes, whereas the genes coding for most eukaryotic phosphatases have not been yet identified, and their cofactors are largely unknown. This observation does not change the outcome of the study, because even if heterotrophic bacteria account for a large portion of the P cycling the conclusions are still valid from a community point of view. But I strongly suggest the authors include this discussion in the paper. It is also possible that the phosphatases of phytoplankton inhabiting this area of the ocean use Fe as cofactor, but as I said the cofactors for eukaryotes are largely unknown.

We again thank the reviewer for their recommendation. Following their advice we have added text regarding the uncertain nature of specific alkaline phosphatase enzymes employed by eukaryotic phytoplankton.

Starting line 29, the revised manuscript now reads:

“As the DOP pool can be orders of magnitude larger than that of DIP in surface ocean waters^{2,8}, the factors regulating access to this pool are likely important for controlling marine primary production. Whilst the identity and prevalence of eukaryotic alkaline phosphatases (APases) remain poorly resolved^{9,10}, analyses of ocean metagenomic datasets have suggested that the phosphate monoesterases comprising the PhoA, PhoX, and PhoD families constitute the dominant bacterial phosphatases in the ocean^{11,12}.”

Specific comments:

Line 10 and line 68: I would erase the “fixed” and just say limitation by N. I find the word “fixed” confusing. Same in line 140.

We have made these modifications to the revised manuscript.

Line 95: in line with what I mentioned before, PhoX and PhoD are abundant and widespread yes, but in bacteria. This is an important piece of information.

We have now made this amendment. The sentence in the revised manuscript now reads:

“Firstly, these experiments have demonstrated the potential for a biogeochemical dependence of P acquisition on Fe availability, with the underlying mechanistic link

presumably being the requirement for Fe in activating the most widespread bacterial APase enzymes in the ocean, PhoX and PhoD^{11,12,19,20}.”

Lines 99-101. Zn did not have an effect from a community point of view, but it did have in Synechococcus cells. Some Synechococcus strains harbour both PhoX and PhoA (Cox and Saito 2013), so maybe that's the reason that both Zn and Fe stimulated Synechococcus cells (e.g Figure S1 exp 2 and exp 5).

As the Reviewer suggests, this result may represent some additional zinc requirement for Synechococcus (possibly PhoA). However, it is difficult to reconcile this with the APA observations and therefore do not provide any new discussion of this in the revised manuscript.

Figure S4. This figure is not mentioned in the text, and does not add any important information.

The Reviewer was correct that this was not mentioned in the initial manuscript. Reviewer 1 also noted this, and in the revised manuscript we have now included a reference to this supplementary figure in the main text.

Starting on line 83 of the revised manuscript the amended sentence is:

“These community-level changes in chlorophyll-a were also largely reflected in cell counts and cellular fluorescence of *Prochlorococcus*, *Synechococcus* and photosynthetic eukaryotes (Supplementary Figs 1-4).”

Although we agree the figure does not add any significant information for this study, it does illustrate how chlorophyll-a per cell changed, matching the physiological responses previously measured in the region (Moore et al., 2008; Davey et al., 2008). As this is in the Supplementary Information and might potentially still be of value for readers comparing the results with previous work we have left the figure in the revised manuscript.

Figure S5. I think this figure should be in the main text.

Whilst respecting the Reviewers advice, we have decided to leave this figure in the supplementary information (now Supplementary Fig. 6). The figure offers an illustrative conceptual overview that might be useful for some readers, but contains no additional information pertinent to the conclusions of the study that are not discussed in the main text of the manuscript.

References

- Buck, K. N., Sohst, B. & Sedwick, P. N. The organic complexation of dissolved iron along the U.S. GEOTRACES (GA03) North Atlantic Section. *Deep. Res. Part II Top. Stud. Oceanogr.* **116**, 152–165 (2015).
- Conway, T. M. & John, S. G. Quantification of dissolved iron sources to the North Atlantic Ocean. *Nature* **511**, 212–215 (2014).
- Davey, M. *et al.* Nutrient limitation of picophytoplankton photosynthesis and growth in the tropical North Atlantic. *Limnol. Oceanogr.* **53**, 1722–1733 (2008).
- Deutsch, C., Sarmiento, J. L., Sigman, D. M., Gruber, N. & Dunne, J. P. Spatial coupling of nitrogen inputs and losses in the ocean. *Nature* **445**, 163–167 (2007).
- Eder, S., Shi, L., Jensen, K., Yamane, K. & Hulett, F. M. A *Bacillus subtilis* secreted phosphodiesterase/alkaline phosphatase is the product of a Pho regulon gene, phoD. *Microbiology* **142**, 2041–2047 (1996).
- Hallmann, A. Enzymes in the Extracellular Matrix of *Volvox*: an Inducible, Calcium-dependent Phosphatase with a Modular Composition. *J. Biol. Chem.* **274**, 1691–1697 (1999).
- Kageyama, H. *et al.* An alkaline phosphatase/phosphodiesterase, PhoD, induced by salt stress and secreted out of the cells of *Aphanothece halophytica*, a halotolerant cyanobacterium. *Appl. Environ. Microbiol.* **77**, 5178–5183 (2011).
- Kathuria, S. & Martiny, A. C. Prevalence of a calcium-based alkaline phosphatase associated with the marine cyanobacterium *Prochlorococcus* and other ocean bacteria. *Environ. Microbiol.* **13**, 74–83 (2011).
- Lin, X., Zhang, H., Huang, B. & Lin, S. Alkaline phosphatase gene sequence characteristics and transcriptional regulation by phosphate limitation in *Karenia brevis* (Dinophyceae). *Harmful Algae* **17**, 14–24 (2012).
- Lis, H., Shaked, Y., Kranzler, C., Keren, N. & Morel, F. M. M. Iron bioavailability to phytoplankton: an empirical approach. *ISME J.* **9**, 1003–1013 (2015).
- Luo, H., Benner, R., Long, R. & Hu, J. Subcellular localization of marine bacterial alkaline phosphatases. *Proc. Natl. Acad. Sci. U. S. A.* **106**, 21219–21223 (2009).
- Mahaffey, C., Reynolds, S., Davis, C. E., Lohan, M. C. & Lomas, M. W. Alkaline phosphatase activity in the subtropical ocean: insights from nutrient, dust and trace metal addition experiments. *Front. Mar. Sci.* **1**, 1–13 (2014).
- Martinez, J. & Azam, F. Periplasmic aminopeptidase and alkaline phosphatase activities in a marine bacterium: implications for substrate processing in the sea. *Mar. Ecol. Prog. Ser.* **92**, 89–97 (1993).
- Mills, M. M., Ridame, C., Davey, M., La Roche, J. & Geider, R. J. Iron and phosphorus co-limit nitrogen fixation in the eastern tropical North Atlantic. *Nature* **429**, 232–292 (2004).
- Mills, M. M. *et al.* Nitrogen and phosphorous co-limitation of bacterial productivity and growth in the oligotrophic subtropical North Atlantic. *Limnol. Oceanogr.* **53**, 1–13 (2008).
- Moore, C. M. *et al.* Relative influence of nitrogen and phosphorus availability on phytoplankton physiology and productivity in the oligotrophic sub-tropical North Atlantic Ocean. *Limnol. Oceanogr.* **53**, 824–834 (2008).

- Moore, C. M. *et al.* Large-scale distribution of Atlantic nitrogen fixation controlled by iron availability. *Nat. Geosci.* **2**, 867–871 (2009).
- Moseley, J. L., Chang, C. W. & Grossman, A. R. Genome-based approaches to understanding phosphorus deprivation responses and PSR1 control in *Chlamydomonas reinhardtii*. *Eukaryot. Cell* **5**, 26–44 (2006).
- Orchard, E. D., Webb, E. A. & Dyhrman, S. T. Molecular analysis of the phosphorus starvation response in *trichodesmium* spp. *Environ. Microbiol.* **11**, 2400–2411 (2009).
- Quisel, J. D., Wykoff, D. D. & Grossman, A. R. Biochemical characterization of the extracellular phosphatases produced by phosphorus-deprived *Chlamydomonas reinhardtii*. *Plant Physiol.* **111**, 839–848 (1996).
- Rijkenberg, M. J. A. *et al.* Fluxes and distribution of dissolved iron in the eastern (sub-) tropical North Atlantic Ocean. *Global Biogeochem. Cycles* **26**, (2012).
- Rijkenberg, M. J. A. *et al.* The distribution of dissolved iron in the West Atlantic Ocean. *PLoS One* **9**, 1–14 (2014).
- Rusch, D. B. *et al.* The Sorcerer II Global Ocean Sampling expedition: Northwest Atlantic through eastern tropical Pacific. *PLoS Biol.* **5**, 0398–0431 (2007).
- Sebastian, M. & Ammerman, J. W. The alkaline phosphatase PhoX is more widely distributed in marine bacteria than the classical PhoA. *ISME J.* **3**, 563–572 (2009).
- Shaked, Y. & Lis, H. Disassembling iron availability to phytoplankton. *Front. Microbiol.* **3**, 1–26 (2012).
- Wu, J. R. *et al.* Cloning of the gene and characterization of the enzymatic properties of the monomeric alkaline phosphatase (PhoX) from *Pasteurella multocida* strain X-73. *FEMS Microbiol. Lett.* **267**, 113–120 (2007).

Reviewers' Comments:

Reviewer #1 (Remarks to the Author)

I am now satisfied with the authors' responses to my initial review comments, and the changes that they have made to the manuscript. I think this revised version is suitable for publication.

Reviewer #2 (Remarks to the Author)

The authors have addressed my comments thoroughly and I recommend publication.

Reviewer #3 (Remarks to the Author)

Browning et al. have addressed most of the points raised by the reviewers, however I still have some additional comments:

- Although the authors now mention that PhoX and PhoD have been mostly described in bacteria, since the authors use Chla for their study I think it would be interesting to note that despite the identity and prevalence of APases in eukaryotic phytoplankton remain poorly resolved, recent studies on some newly identified APases suggest that they are calcium-based enzymes, like PhoX and PhoD, and not Zn based (Lin et al 2013. Identification and characterization of an extracellular alkaline phosphatase in the marine diatom *Phaeodactylum tricornutum*. *Mar Biotechnol*, and Lin X, Wang L, Shi X, Lin S. (2015). Rapidly diverging evolution of an atypical alkaline phosphatase (PhoAaty) in marine phytoplankton: Insights from dinoflagellate alkaline phosphatases. *Front Microbiol* 6. doi:10.3389/fmicb.2015.00868). These observations suggest that the prevalence of calcium-based enzymes (reported in Kathuria et al.) might be extendable to eukaryotic phytoplankton. However the use of Fe as a cofactor for these enzymes needs to be further examined.

- In the revised version of the ms there is quite a lot of discussion about cytoplasmic PhoA, and the potential underestimation of PhoA-based activity using MUF-P (line 107-109, and sup. Discussion). I strongly disagree with this idea. *E. coli* PhoA and all the Pho-regulated PhoA (enzymes induced upon phosphate stress) that have been described so far are periplasmic or membrane bound enzymes. It makes sense because alkaline phosphatases are promiscuous enzymes that hydrolyse a broad spectrum of phosphate monoesters. Cytoplasmic enzymes require that the substrate goes inside the cells, and only very small phosphorylated molecules can go through the cell membrane. There are some specific membrane transporters for low molecular weight phosphorylated compounds, like glycerol phosphate (Luo et al., 2009). But it is unlikely that marine bacteria can directly take up the vast majority of phosphoesters, because direct uptake of these compounds would rely on the existence of specific membrane transporters. The synthesis of specific transporters for this diverse pool of compounds is costly for the cell, whereas the existence of a single periplasmic or extracellular protein with broad substrate specificity would enable access to a wide variety of compounds, minimizing the energy expense. PhoA is a very broad family of proteins and it is likely that these cytoplasmic proteins described in Luo et al. 2009 are related to internal P metabolism, but not involved in the acquisition of P from dissolved P compounds in the environment.

Therefore, based on all these comments I find the MUF-P assay is perfectly valid for addressing the effect of iron in P acquisition from monophosphate esters. Although I am aware that most of this discussion appears now in the supplementary material, I think it is important to note that cytoplasmic enzymes likely play a minor role in the utilization of organic P compounds.

Minor comments: references need to be properly formatted in the text because now they appear sometimes as `ref. 16`, etc.

Responses to Reviewer 3's comments

Manuscript: 'Iron limitation of microbial phosphorus acquisition in the tropical North Atlantic' by T.J. Browning et al.

Responses to comments are in bold.

Reviewer #3 (Remarks to the Author):

Browning et al. have addressed most of the points raised by the reviewers, however I still have some additional comments:

We thank Reviewer 3 for taking the time to make additional comments on our revised manuscript and address these below.

Although the authors now mention that PhoX and PhoD have been mostly described in bacteria, since the authors use Chla for their study I think it would be interesting to note that despite the identity and prevalence of APases in eukaryotic phytoplankton remain poorly resolved, recent studies on some newly identified APases suggest that they are calcium-based enzymes, like PhoX and PhoD, and not Zn based (Lin et al 2013). Identification and characterization of an extracellular alkaline phosphatase in the marine diatom Phaeodactylum tricornutum. Mar Biotechnol, and Lin X, Wang L, Shi X, Lin S. (2015). Rapidly diverging evolution of an atypical alkaline phosphatase (PhoAaty) in marine phytoplankton: Insights from dinoflagellate alkaline phosphatases. Front Microbiol 6. doi:10.3389/fmicb.2015.00868). These observations suggest that the prevalence of calcium-based enzymes (reported in Kathuria et al.) might be extendable to eukaryotic phytoplankton. However the use of Fe as a cofactor for these enzymes needs to be further examined.

We have now added the two suggested references to the manuscript (line 32). However, as the reviewer notes, the metal content of these newly identified eukaryotic APases are still uncharacterized. Therefore whilst it is an interesting point that these candidate Fe-containing APases could have the potential to have partially contributed to our observed community-level responses to added Fe, it is perhaps currently premature for us to speculate. Therefore we have kept our discussion limited to those enzymes both currently identified as most common in the ocean and robustly associated with Fe or Zn.

In the revised version of the ms there is quite a lot of discussion about cytoplasmic PhoA, and the potential underestimation of PhoA-based activity using MUF-P (line 107-109, and sup. Discussion). I strongly disagree with this idea. E.coli PhoA and all the Pho-regulated PhoA (enzymes induced upon phosphate stress) that have been described so far are periplasmic or membrane bound enzymes. It makes sense because alkaline phosphatases are promiscuous enzymes that hydrolyse a broad spectrum of phosphate monoesters. Cytoplasmic enzymes require that the substrate goes inside the cells, and only very small phosphorylated molecules can go through the cell membrane. There are

some specific membrane transporters for low molecular weight phosphorylated compounds, like glycerol phosphate (Luo et al., 2009). But it is unlikely that marine bacteria can directly take up the vast majority of phosphoesters, because direct uptake of these compounds would rely on the existence of specific membrane transporters. The synthesis of specific transporters for this diverse pool of compounds is costly for the cell, whereas the existence of a single periplasmic or extracellular protein with broad substrate specificity would enable access to a wide variety of compounds, minimizing the energy expense. PhoA is a very broad family of proteins and it is likely that these cytoplasmic proteins described in Luo et al. 2009 are related to internal P metabolism, but not involved in the acquisition of P from dissolved P compounds in the environment. Therefore, based on all these comments I find the MUF-P assay is perfectly valid for addressing the effect of iron in P acquisition from monophosphate esters. Although I am aware that most of this discussion appears now in the supplementary material, I think it is important to note that cytoplasmic enzymes likely play a minor role in the utilization of organic P compounds.

In response to one comment from Reviewer 2 we included a caveat—briefly referred to in the main text of the revised manuscript (lines 107–109) and discussed at greater length in the Supplementary Information—that our MUF-P assay may have the potential to be biased at some level by the possible partial location of certain forms of PhoA in the cytoplasm. In contrast to Reviewer 2, Reviewer 3 here disagrees with the potential for a cytoplasmic localized PhoA acting upon externally-derived DOP, thereby suggesting that our assay would be equally sensitive to changes in both Zn (PhoA) and Fe (PhoX) related enzymatic activities.

We actually largely agree with Reviewer 3’s comments; indeed, in our original response to Reviewer 2 we made similar arguments and the revised Supplementary Information included a comment with respect to internal cycling roles of any cytoplasmic PhoA (SI line 9). However, in response to Reviewer 3’s comments above we have made a few additional changes in order to clarify our perspective within the discussion in the Supplementary Information, as well as simplifying this discussion in places. We feel this discussion represents a fair summary of the current state of the literature and a good compromise given the divergent views/requests of Reviewers 2 and 3 with respect to this particular point. Moreover and most importantly, we reiterate that our primary conclusion relating to Fe limitation of microbial P acquisition remains insensitive to this debate.

Minor comments: references need to be properly formatted in the text because now they appear sometimes as ‘ref. 16’, etc.

As the reviewer notes we altered some of the formatting of our references, for example:

“Until recently, cofactor requirements were understood to be two zinc (Zn) and one magnesium (Mg) ions for PhoA (ref. 16)...”

However, these formatting changes are restricted to cases where there was potential for the superscript reference to cause confusion; in this specific case a superscript number attached to “PhoA” might be mistaken to represent part of our description of “PhoA” rather than identifying a reference. As far as we are aware this notation is the preferred format for this journal, but this will of course be changed if requested by the Editor.

We thank Reviewer 3 again for their additional comments on our revised manuscript.

Reviewers' Comments:

Reviewer #3:

Remarks to the Author:

The authors have addressed my comments and I recommend publication.